# Leveraging Machine Learning and Simulation to Advance Disaster Preparedness Assessments through FEMA National Household Survey Data

Zhenlong Jiang [1] , Yudi Chen [2], Ting-Yeh Yang [3], Wenying Ji [4], Zhijie (Sasha) Dong [5] and Ran Ji [1,*]

1 Department of Systems Engineering and Operations Research, George Mason University, Fairfax, VA 22030, USA
2 Department of Management Science and Engineering, College of Economics and Management, Nanjing University of Aeronautics and Astronautics, Nanjing 211106, China
3 College of Engineering and Computing, George Mason University, Fairfax, VA 22030, USA
4 Department of Civil, Environmental, and Infrastructure Engineering, George Mason University, Fairfax, VA 22030, USA
5 Department of Construction Management, University of Houston, Houston, TX 77204, USA
* Correspondence: rji2@gmu.edu; Tel.: +1-703-993-5993

**Abstract:** Effective household and individual disaster preparedness can minimize physical harm and property damage during catastrophic events. To assess the risk and vulnerability of affected areas, it is crucial for relief agencies to understand the level of public preparedness. Traditionally, government agencies have employed nationwide random telephone surveys to gauge the public's attitudes and actions towards disaster preparedness. However, these surveys may lack generalizability in certain affected locations due to low response rates or areas not covered by the survey. To address this issue and enhance the comprehensiveness of disaster preparedness assessments, we develop a framework that seamlessly integrates machine learning and simulation. Our approach leverages machine learning algorithms to establish relationships between public attitudes towards disaster preparedness and demographic characteristics. Using Monte Carlo simulation, we generate datasets that incorporate demographic information of the affected location based on government-provided demographic distribution data. The generated dataset is then input into the machine learning model to predict the disaster preparedness attitudes of the affected population. We demonstrate the effectiveness of our framework by applying it to Miami-Dade County, where it accurately predicts the level of disaster preparedness. With this innovative approach, relief agencies can have a clearer and more comprehensive understanding of public disaster preparedness.

**Keywords:** disaster management; household preparedness prediction; machine learning; Monte Carlo simulation

## 1. Introduction

Natural disasters pose significant threats to the lives and property of those affected, frequently leading to shortages of essential resources [1–3]. However, it is important to note that most fatalities and damages caused by disasters are preventable [4]. Adequate household disaster preparedness can significantly reduce the negative consequences of disasters and ensure that individuals can care for themselves and their families during the first 72 h following a disaster [5].

The US government estimates that investing one dollar in pre-disaster mitigation and preparedness efforts can prevent up to six dollars in losses from potential disasters [6]. This is why most governmental relief agencies encourage the public to conduct disaster preparedness to hedge against the risk of natural disasters. For instance, the Federal Emergency Management Agency (FEMA) launched a campaign called *Get Ready* in 2003 to encourage people to prepare for emergencies and disasters. The campaign promotes the

message to *get a kit, make a plan, and be informed* about different types of emergencies and their appropriate responses.

For disaster relief agencies, gaining knowledge about public disaster preparedness is essential for disaster risk and vulnerability assessments [7]. The survey is the most effective method to discover public attitudes and actions about disaster preparedness. Many relief agencies conduct surveys to learn public attitudes and actions about disaster preparedness. For example, FEMA has conducted the National Household Survey (NHS) annually to track the progress of household disaster preparedness via phone interviews since 2013 [8]. Moreover, in academia, many researchers conduct surveys to analyze public attitudes about disaster preparedness in many countries and cities such as Japan [9], Serbia [10], Istanbul [11], and Tehran [12].

Conventional methods of surveying public disaster preparedness rely on either telephone surveys or convenience samples, both of which suffer from limitations. Telephone surveys are notorious for low response rates, while convenience samples have restricted generalizability. A case in point is the 2019 National Household Survey (NHS) conducted by [13] in the United States, which sampled 5025 households, including an oversampling of 510 interviewees from high-risk hurricane areas. Despite the focus on high-risk areas by the Federal Emergency Management Agency (FEMA), the 2019 NHS data show only 78 interviewees residing in Georgia, a state particularly vulnerable to hurricane damage. The limited sample size for Georgia renders the survey results unrepresentative of the actual state of household disaster preparedness, potentially leading decision-makers to miscalculate the population's vulnerability and overlook the needs of those most at risk. Therefore, more comprehensive and reliable methods of gauging public disaster preparedness are needed to enable better-informed relief efforts.

In academia, researchers often utilize statistical analysis techniques such as regression and hypothesis testing on historical survey data to address the generalizability issue of traditional surveys. Identifying the features that impact public attitudes towards disaster preparedness can guide disaster relief agencies in quickly identifying vulnerable groups and implementing targeted relief efforts when disasters occur [14]. For example, ref. [15] utilized linear regression to analyze survey data from South Carolina and found that age and income of surveyed residents had a significant correlation with disaster preparedness attitudes. Similarly, ref. [16] developed a logistic regression model based on surveys conducted in the New Orleans metro area and Los Angeles County. Their model demonstrated that residents' confidence in government disaster management abilities and access to disaster preparedness information were key factors that could enhance the level of disaster preparedness.

Several studies have identified factors that impact disaster preparedness attitudes among the public. The studies of [17–20] found that females and individuals with higher levels of education were more likely to engage in disaster preparedness measures for residents in Komoro, San Diego, Taiwan, and rural villages in northwest China, respectively. In Istanbul, higher education levels, earthquake experience, living in higher-earthquake-risk areas, home ownership, and being between 35 and 54 years old were found to be positively correlated with disaster preparedness [11]. Similarly, in Florida, age, home ownership, house type, income, and race were found to have a significant impact on attitudes towards hurricane preparedness [14]. In Iran, income level, disaster experience, living area, and occupation were found to significantly affect preparedness scores [12]. Education level and disaster experience were found to strongly impact disaster preparedness in Thailand and the Philippines [21]. In the US, several factors such as informal support, income level, health status, and disaster experience have been found to be correlated with disaster preparedness among elders, as reported by [22]. Ref. [23] also obtained similar results based on a survey conducted in the US, where they found that age, income, and gender are highly related to the public attitudes towards disaster preparedness. Similarly, it has been discovered that disaster experience has a positive impact on the disaster preparedness of residents living in Japan, as reported by [9]. In the Slovak Republic, a survey conducted by [24] found that individuals with prior disaster experience and higher income levels were

more likely to engage in disaster preparedness, as determined through linear and logistic regression analyses.

The impact of ethnicity on disaster preparedness has also been studied. Hispanics in the US were found to be less likely to conduct disaster preparedness, while income level, education, age, and disaster experience were positively related to preparedness [25]. Gender, ethnic group, age, medical conditions, healthcare access, number of children, income level, and evacuation experience were also found to significantly impact disaster preparedness attitudes among interviewees [26]. Lastly, gaining preparedness information or having disaster experience were found to increase the likelihood of disaster preparedness, while living in a rental house or being Latino or Asian were negatively correlated with disaster preparedness [27]. The summarized information of studies from the literature review is presented in Table 1.

**Table 1.** Summary of the reviewed literature.

| Paper | Regions | Year | Obser. | Meth. | DE | IA | A | I | G | SL | R | MS | C | EL | HL | HO | HT | JT |
|---|---|---|---|---|---|---|---|---|---|---|---|---|---|---|---|---|---|---|
| [15] | SC | 1993 | 257 | Lin. | | | ✓ * | ✓ * | ✓ | | | | | | | | | |
| [16] | NO, LA | 1999 | 404 | Log. | ✓ | ✓ * | ✓ | ✓ | ✓ | | ✓ | ✓ | ✓ | ✓ | | | | |
| [23] | USA | 2006 | 1629 | Lin. | | | ✓ * | ✓ * | ✓ * | ✓ | | | | ✓ | ✓ | | | |
| [11] | Istanbul | 2007 | 1123 | Log. | ✓ * | ✓ * | ✓ * | ✓ | ✓ | | | ✓ | ✓ | ✓ * | ✓ * | ✓ * | ✓ | |
| [14] | Florida | 2006 | 1200 | Chi. | | | ✓ * | ✓ * | ✓ | | ✓ * | | ✓ * | ✓ | ✓ | ✓ * | ✓ * | |
| [17] | Komoro | 2011 | 4000 | Log. | | | ✓ * | | ✓ * | | | | ✓ | ✓ * | | | | ✓ |
| [12] | Iran | 2014 | 1250 | Lin. | ✓ * | | | ✓ * | ✓ | | | | ✓ | ✓ * | ✓ | ✓ | ✓ | ✓ * |
| [18] | SD | 2016 | 983 | Log. | | | ✓ | ✓ * | ✓ * | | | ✓ * | ✓ | ✓ * | | | | ✓ |
| [19] | Taiwan | 2013 | 4082 | Log. | ✓ * | | ✓ | ✓ | ✓ | | | | ✓ | ✓ * | | ✓ | | |
| [21] | SAC | 2014 | 2199 | Log. | ✓ * | | ✓ | ✓ | ✓ | | | ✓ | ✓ * | ✓ * | ✓ * | | ✓ * | ✓ |
| [22] | US | 2014 | 719 | Log. | | | ✓ | ✓ * | ✓ | | ✓ | | ✓ | ✓ | | | | |
| [9] | Japan | 2013 | 20,726 | Lin. | ✓ * | | ✓ * | ✓ * | ✓ * | | | | ✓ * | ✓ * | ✓ * | | ✓ * | |
| [28] | USA | 2010 | 1711 | Lin. | | | ✓ * | ✓ * | ✓ | | ✓ * | | | ✓ | | | | |
| [25] | RGV | 2014 | 740 | Lin. | ✓ * | | ✓ * | ✓ * | ✓ | | ✓ | | | ✓ | | | | |
| [10] | Serbia | 2015 | 2500 | Lin. | | | ✓ | ✓ | ✓ * | | | ✓ * | | ✓ * | | | | |
| [29] | USA | 2008 | 1137 | Log. | | | ✓ * | ✓ * | ✓ | | ✓ | ✓ | | ✓ * | ✓ | | ✓ | |
| [20] | China | 2015 | 3245 | Lin. & Log. | | | ✓ | ✓ | ✓ * | | | | | ✓ * | ✓ * | | ✓ | |
| [26] | RGV. | 2017 | 590 | Chi. | ✓ * | | ✓ * | ✓ * | ✓ * | | ✓ * | | ✓ * | ✓ | | | ✓ | ✓ |
| [24] | SR | 2019 | 794 | Lin. & Log. | ✓ * | | ✓ | ✓ * | | | | | | ✓ | | ✓ | | |
| [27] | USA | 2018 | 5003 | Log. | ✓ * | ✓ * | ✓ | ✓ | ✓ | ✓ | ✓ * | | | ✓ | | ✓ * | | |
| This paper | GC | 17,18 | 1604 | ML | ✓ | ✓ | ✓ | ✓ | ✓ | | ✓ | | | ✓ | | | | |

Meth.—Methodology; Lin.—Linear regression; Log.—Logistic regression; Chi.—Chi-Square test; SC—South Carolina; NO—New Orleans; LA—Los Angeles; SD—San Diego, CA; SAC: Southeast Asia Countries, Thailand and Philippines; GC—Gulf Coast, US; RGV—Rio Grande Valley, Texas; SR—Slovak Republic; RVNC—Rural villages in northwest China; DE—Disaster experience; IA—Information accessibility; I—Income; A—Age; G—Gender; R—Race; SL—Speaking language; MS—Marital status; C—Children in home; EL—Education level; HL—Home location; HO—Home ownership; HT—House type; JT—Job type. ✓—Analyzed independent variable; ✓ *—Significant independent variable.

While many factors have been found to be significant in shaping public attitudes toward disaster preparedness, this information alone is insufficient for relief agencies to address the diversity and complexity of the affected population. For instance, ethnicity and income level have been found to be positively related to disaster preparedness for those living in the Rio Grande Valley [25], but it is difficult to determine which community is most vulnerable among multiple communities with different ethnicities and income levels. Additionally, previous studies have focused on testing significance without evaluating model performance, such as prediction accuracy, leading to limited predictive capabilities. To address this gap, we apply multiple machine learning algorithms to FEMA NHS data to develop a more accurate and comprehensive estimation of residents' attitudes towards disaster preparedness. To address the generalizability challenge, we utilize the Monte Carlo simulation approach to simulate residents' demographic features and use this data to train ML models to predict disaster preparedness in affected locations.

A relevant study for our research is [27]. In their study, the authors utilized logistic regression to analyze FEMA's 2018 NHS data, examining the association between surveyed factors and disaster preparedness attitudes among US households. However, the authors

did not remove invalid answers such as "do not know" or "reject to answer", which may have affected their model's performance, and consequently, the analysis results and insights. Therefore, we provide a detailed step-by-step data cleaning strategy by carefully examining the missing data situation. Additionally, the authors did not explore other machine learning algorithms, which could have provided further insights into disaster preparedness analysis. Thus, we experimented with various machine learning algorithms, including Logistic Regression (LR), Random Forest (RF), eXtreme Gradient Boosting (XGBoost), Support Vector Machine (SVM), K-Nearest Neighbors (KNN), and Artificial Neural Network (ANN). It is also important to note that in [27], the authors focused on characterizing the relationship between the disaster preparedness and various factors, but they did not utilize historical NHS data to predict disaster preparedness levels in affected locations, marking a significant difference between their research and ours.

In this paper, we developed an integrated machine learning and Monte Carlo simulation framework to aid relief agencies in predicting the disaster preparedness of affected areas. Incorporating the Monte Carlo simulation into our proposed framework offers a flexible and adaptable solution for acquiring demographic data for locations that are not covered by surveys, or where survey data may be sparse or incomplete. This innovative method takes into account the real-world variability and uncertainties present in the affected population, generating a more comprehensive understanding of their characteristics and preparedness levels. The fusion of machine learning and Monte Carlo simulation enables a more detailed and complete analysis of the vulnerability of affected areas, allowing relief agencies to better plan and allocate resources during catastrophic events.

Our paper makes contributions to the literature from the following three perspectives:

- Our study presents a novel approach to disaster preparedness analysis by developing machine learning models that accurately predict household attitudes towards disaster preparedness. To our knowledge, this is the first time that machine learning algorithms such as XGBoost and artificial neural networks have been applied to this type of analysis.
- We integrate the Monte Carlo simulation with the machine learning framework, utilizing simulated demographic data derived from US census data as input for the machine learning models to predict disaster preparedness levels in targeted locations. This method proves especially beneficial in instances where surveys have not covered affected areas, or when data in the affected area are scarce. The inclusion of simulation significantly enhances the generalizability of the proposed framework, making it more adaptable for a variety of disaster preparedness assessment scenarios.
- We further conducted sensitivity analysis to evaluate the impact of information awareness on disaster preparedness. The results show that higher information awareness improves disaster preparedness levels, which can help the relief agencies evaluate the effectiveness of their education and communication efforts in increasing public information awareness about disaster preparedness.

The remainder of this paper is organized as follows. Section 2 presents the proposed methodology for predicting the level of disaster preparedness. In Section 3, we present the results of extensive numerical tests to evaluate the efficiency of our proposed solution framework and assess the model's performance. We conclude the paper in Section 4 and suggest future research directions.

## 2. Methodology

The proposed framework is summarized in Figure 1. There are four modules in the proposed framework including (1) Data Collection and Cleaning, (2) Model Training and Selection, (3) Residents' Feature Simulation, and (4) Preparedness Level Prediction, which will be introduced in detail in the following subsections.

**Figure 1.** Flowchart of proposed framework.

## 2.1. Data Collection and Cleaning

FEMA administers the National Household Survey (NHS) to investigate the American public's progress in personal disaster preparedness through telephone interviews. The survey consists of three major components: attitudes toward hazard preparedness, hazard experience, and demographic information. The interviewee's attitude about disaster preparedness behavior is classified into five levels: Pre-contemplation, Contemplation, Preparation, Action, and Maintenance. In our study, we consider individuals in the Action and Maintenance levels to have prepared for disasters, while those in the Pre-contemplation, Contemplation, and Preparation levels are deemed unprepared. We thus convert the multi-level categorical variable into a binary one (i.e., prepared vs. unprepared) and use it as the dependent variable in our predictive modeling analysis.

We retrieved the 2017, 2018, and 2019 FEMA NHS records from the official website [13,30,31] and manually selected seven states (Florida, Texas, Louisiana, South Carolina, Alabama, Georgia, and Mississippi) that were affected by hurricanes between 2015 and 2020 [32] for analysis. To construct the machine learning models, we selected demographic features, including age, income, gender, education level, ethnicity, disaster experience, and awareness of information, based on previous studies that established a significant relationship between these variables and disaster preparedness [9,16,33–35] and the availability of NHS data. In addition, we kept the state name and zip codes from the raw datasets to identify the interviewee's address and evaluate the simulation results. The training dataset comprised NHS 2017 and 2018, and NHS 2019 served as the testing dataset to assess the model's performance and the predicted level of disaster preparedness accuracy. We removed observations with invalid information caused by the answer "Refused" or "Do not know" and matched the interviewees' zip code location with their state name; any unmatched observations were removed. The variables used in the study and their data types are summarized in Table 2, with the dependent variable referred to as Preparedness for brevity. Disaster experience, awareness of information, and education level are denoted as Experience, Information, and Education, respectively.

**Table 2.** Summary of variables for model training.

| Variable | Definition of Variable | Data Type |
|---|---|---|
| Preparedness | Whether the household prepare for disaster | Binary |
| Experience | Whether the household experience a disaster | Binary |
| Information | Whether the household received disaster prevention information | Binary |
| Age | Demographic feature: interviewee's age | Integer |
| Income | Demographic feature: interviewee's income per year | Factor |
| Gender | Demographic feature: interviewee's gender | Binary |
| Education | Demographic feature: interviewee's levels of educational attainment | Factor |
| Ethnicity | Demographic feature: interviewee's ethnicity | Factor |

## 2.2. Model Training and Selection

In this study, we utilized six different machine learning models, namely Logistic Regression (LR), Random Forest (RF), eXtreme Gradient Boosting (XGBoost), Support Vector Machine (SVM), K-Nearest Neighbors (KNN), and Artificial Neural Network (ANN), to examine the relationship between public attitudes about disaster preparedness and the introduced variables. Logistic Regression is a linear model that is commonly used to predict binary outcomes, while Random Forest is an ensemble model that combines multiple decision trees to improve prediction performance. XGBoost is a gradient boosting model that utilizes decision trees and is particularly effective for large datasets. SVM

is a binary classification model that seeks to find the best hyperplane that separates the different classes of data, while KNN is a non-parametric model that makes predictions based on the nearest neighbors of a given data point. Finally, Artificial Neural Network is a set of interconnected nodes organized into layers that can be used to make complex predictions. The choice of which machine learning model to employ depends on a variety of factors, including prediction accuracy and interpretability. Interested readers can refer to the classical machine learning book by [36] for more detailed descriptions of each of these models.

The module selection process begins with data pre-processing to transform predictor variables, such as Income, Education, and Ethnicity into binary dummy variables. The variable Age is re-scaled using min–max normalization to convert it between 0 and 1 with the following equation:

$$\text{Age}^* = \frac{\text{Age} - \min(\text{Age})}{\max(\text{Age}) - \min(\text{Age})}$$

when training the ML model, where a grid search approach and *K*-Fold Cross Validation (CV) are used to optimize hyperparameters. In the CV process, the training data are randomly and evenly divided into *K* folds, and the model is trained on $K - 1$ folds and tested on the remaining fold. This process is repeated K times to obtain an average accuracy. In our study, we use ten folds. The optimal parameter setting is obtained by comparing the CV accuracy obtained on each parameter grid.

The final step is model validation, where a hold-out test is used to evaluate model performance. We obtain the predicted preparedness attitude of each household based on the 2019 NHS data and use Accuracy, AUC, F-1 score, and Specificity to evaluate model performance. More technical details about model training will be illustrated in Section 3.

### 2.3. Residents' Feature Simulation

Monte Carlo simulation is a widely used technique in many fields, including engineering, physics, economics, and finance, to estimate the distribution of an uncertain variable [37]. In essence, Monte Carlo simulation involves generating a large number of data points, usually following a probabilistic distribution, to simulate a real-world scenario. The generated data can then be used to assess the likelihood of certain outcomes or to inform decision-making processes. In the context of disaster preparedness, Monte Carlo simulation can be used to generate demographic profiles of residents living in an affected location. This is particularly useful when survey data (e.g., NHS data) on the demographics of the affected population are limited or not available.

In this module, we utilize the Monte Carlo simulation approach to simulate data points for selected demographic features of residents in the affected county. To accomplish this, we first collect distribution information about residents' demographic features, such as age, income, education level, ethnicity, and gender, from the United States Census Bureau (USCB). To perform Monte Carlo simulation, a probabilistic distribution is selected based on available data or expert knowledge. For example, the distribution of age can be estimated from the USCB data and represented as a normal distribution. The simulation then generates a large number of data points, typically using a random number generator, that follow the selected distribution.

Additionally, we denote the "Experience" and "Information" as two variables following binary probability distributions, whose expected probability is calculated as the percentage of residents who have disaster experience and those who are aware of disaster preparedness information, respectively, based on the 2017 and 2018 NHS data. Once a sufficient number of data points have been generated, they can be combined to create a demographic profile of a resident living in the affected location. This process is repeated multiple times, resulting in various simulated data of demographic profiles of residents with disaster-related experience and information awareness, which can be used as a good representation of the residents in affected areas with limited data. This approach helps

ensure the generalizability of the proposed framework by accounting for the variability and uncertainty in the demographic features of residents in the affected areas.

### 2.4. Preparedness Level Prediction

In the Preparedness Level Prediction module, we utilize the top-performing model from the second module to predict the level of disaster preparedness for the impacted area. To do so, we construct data about the residents' information for the affected location using Monte Carlo simulation in the third module. By feeding the simulated dataset as the input data for the selected ML model, we can systematically and accurately determine whether a household in the affected location is engaged in disaster preparedness. By taking an average of each resident's preparedness status, we can calculate the disaster preparedness level (as a percentage of the total number of households) for the affected location.

To ensure robustness, we generate multiple simulated datasets in the third module, enabling us to obtain a disaster preparedness level for each dataset by repeating the above process, and eventually obtain a distribution of preparedness level across all simulated datasets. Finally, we can determine the most preferred disaster preparedness level for the affected location by analyzing a histogram of the obtained disaster preparedness level.

## 3. Results and Discussion

### 3.1. Training Data Set Exploration

Following our data collection schemes introduced in Section 2.1, after performing data cleaning, we obtain 1604 observations in the training dataset (e.g., from the NHS 2017 and 2018 datasets) and 864 observations in the testing dataset (e.g., from the NHS 2019 dataset). Table 3 summarizes some descriptive statistics of variables in the training dataset. From Table 3, we can see that the majority of households (61.16%) have taken some action to prepare for potential disasters. Additionally, a significant proportion of families (72.63%) have experienced a disaster in the past, while only 60.29% have received information about disaster preparedness.

The age range of the interviewees is between 18 and 95 years old, with a mean age of 50.52 and a median age of 51. The income of the households varies widely, with 58% of interviewees earning more than USD 4000 in household income, while 7.23% earn less than USD 1000. Furthermore, most interviewees (72%) hold a college or higher degree. In terms of race, the majority of interviewees are white people (75%), followed by African-Americans (20.26%), and other races make up the remaining 5%. These demographic factors can help us gain preliminary insights into the disaster preparedness levels of households in the affected location.

**Table 3.** Training dataset summary.

| Preparedness | | Income | |
|---|---|---|---|
| No | 38.84% | Below 999 | 7.23% |
| Yes | 61.16% | 1000–3999 | 34.29% |
| **Experience** | | 4000–9999 | 40.03% |
| No | 27.37% | Over 10,000 | 18.45% |
| Yes | 72.63% | **Education** | |
| **Information** | | Less High School | 5.67% |
| No | 39.71% | High School | 17.96% |
| Yes | 60.29% | Vocational School | 5.42% |
| **Age** | | College | 24.31% |
| Min | 18 | College Graduate | 28.74% |
| Mean | 50.52 | Post Graduate | 17.89% |
| Median | 51 | **Ethnicity** | |
| Max | 95 | White | 75.00% |
| **Gender** | | African-American | 20.26% |
| Men | 50.12% | Asian | 2.12% |
| Women | 49.88% | American Indian | 1.68% |
| | | Hawaiian | 0.94% |

*3.2. Model Performance Evaluation*

In this study, we utilized 10-fold cross-validation and accuracy as the evaluation metric to optimize the hyperparameters of our machine learning models. Table S1 in the Supplementary Materials lists the candidate and resulting tuned model parameters. We further evaluated the performance of the model on the testing dataset using additional metrics, including AUC (Area Under the ROC Curve), F-1 score, and Specificity. To be self-contained, we briefly describe each of these metrics below.

1.  Accuracy: Accuracy is a widely used evaluation metric for classification models. It is defined as the proportion of correctly classified instances to the total number of instances. In other words, it measures the overall correctness of a model's predictions. The value of Accuracy is computed as follows:

$$\text{Accuracy} = \frac{\text{TP} + \text{TN}}{\text{TP} + \text{TN} + \text{FP} + \text{FN}}$$

    where TP, TN, FP, and FN represent True Positive, True Negative, False Positive, and False Negative, respectively.

2.  AUC: The Area Under the ROC Curve (AUC) is a performance metric used to evaluate binary classification models at different classification thresholds. ROC stands for Receiver Operating Characteristic. It is a graphical representation of the performance of a binary classification model. The ROC curve plots the true positive rate (TPR) against the false positive rate (FPR) at different classification thresholds. It is a widely used evaluation metric for classification models, particularly in machine learning and statistics. The AUC is computed by plotting the True Positive Rate (TPR) against the False Positive Rate (FPR) at different threshold values and then calculating the area under the resulting curve. The AUC ranges from 0 to 1, with higher values indicating better performance. A model with an AUC of 0.5 is equivalent to random guessing, while a model with an AUC of 1 is perfect. In summary, the AUC is a useful metric for evaluating the overall performance of a binary classification model.

3.  F-1 score: The F-1 score is a popular evaluation metric for binary classification models that combines both precision and recall into a single score. Precision measures the proportion of correctly predicted positive instances out of all predicted positive instances, while recall measures the proportion of correctly predicted positive instances out of all actual positive instances. The harmonic mean of precision and recall is used to calculate the F-1 score, which ranges from 0 to 1, with higher values indicating better performance. A perfect classifier would have an F-1 score of 1, while a completely random classifier would have an F-1 score of 0. The formulation to compute the F-1 score is given below.

$$\text{F-1 score} = 2 \times \frac{\text{precision} \times \text{recall}}{\text{precision} + \text{recall}}$$

    where precision $= \text{TP}/(\text{TP} + \text{FP})$ and recall $= \text{TP}/(\text{TP} + \text{FN})$.

4.  Specificity: Specificity measures the proportion of true negative instances that are correctly identified by the model. It is calculated by dividing the number of true negative instances by the sum of true negative and false positive instances. In other words, specificity measures the model's ability to correctly identify negative instances as negative. A high specificity score indicates that the model is very good at avoiding false positives (i.e., it has a low rate of false alarms) and is useful in applications where the cost of a false positive is high, such as the classification of household preparedness in the disaster context as in our problem.

$$\text{Specificity} = \frac{\text{TN}}{\text{TN} + \text{FP}}$$

Table 4 provides a summary of the training CV accuracy and hold-out testing performance of each of the six proposed ML models, which are evaluated using the above-mentioned metrics. The threshold used to calculate the CV and hold-out test accuracy is 0.5. The bold numbers in the table represent the top three highest values of each evaluation metric.

**Table 4.** Model performance.

|  | Training | Hold-Out Test |  |  |  |
|---|---|---|---|---|---|
|  | Accuracy | Accuracy | AUC | F-1 Score | Specificity |
| LR | **0.716** | **0.704** | **0.741** | **0.769** | 0.770 |
| RF | 0.708 | **0.700** | 0.727 | **0.770** | **0.800** |
| XGBoost | **0.719** | **0.704** | **0.745** | **0.772** | **0.786** |
| SVM | 0.702 | 0.686 | 0.728 | 0.761 | **0.783** |
| KNN | 0.645 | 0.652 | 0.665 | 0.731 | 0.759 |
| ANN | **0.720** | 0.686 | **0.734** | 0.749 | 0.732 |

From the results presented in Table 4, it is evident that LR, XGBoost, and ANN exhibit higher CV accuracy compared to the other three ML models. Among the six models, ANN achieves the highest CV accuracy of 72%, which is 1% and 2% higher than XGBoost and LR, respectively. The CV accuracy of RF and SVM does not exceed 71%, but it is over 70%. In contrast, KNN performs the worst among the six models, with a CV accuracy that does not exceed 65%.

In terms of hold-out testing performance, XGBoost outperforms all other models in all evaluation metrics except specificity. For instance, XGBoost has an AUC value of 0.745, which is 0.004, 0.018, 0.017, 0.08, and 0.011 higher than the AUC value of LR, RF, SVM, KNN, and ANN, respectively. Moreover, XGBoost has the second-highest specificity value, which is 0.014 lower than that of RF. The LR model also performs well in the disaster preparedness prediction hold-out test compared to RF, SVM, KNN, and ANN. Although the specificity value of the LR model is lower than that of RF, XGBoost, and SVM, the prediction accuracy (0.716) and AUC (0.741) of LR are the second-highest among the proposed models.

The RF and ANN models also perform well in some evaluation metrics but do not dominate over LR and XGBoost in all evaluation measures. Overall, XGBoost presents the best performance on all evaluation metrics for the hold-out test, making it the best model among the six. As such, we will employ LR and XGBoost for further analysis in the following sections, given that LR can present a more intuitive relationship between the dependent and independent variables.

To conduct further analysis, we will mainly focus on the results of logistic regression and XGBoost models considering the trade-off between prediction accuracy and interpretability. Logistic regression offers a simple and transparent equation, which can be used to interpret the impact of each predictor on individual attitudes toward disaster preparedness. Although it may not provide the most accurate predictions, it is an effective method to understand the relationship between the predictors and the outcome. On the other hand, XGBoost is a black-box model that provides high prediction accuracy when estimating individual attitudes about disaster preparedness. However, due to the ensemble nature of XGBoost, it is challenging to obtain a transparent interpretation of the relationship between predictors and the outcome variable.

*3.3. Feature Importance and Significance Analysis*

This subsection focuses on the feature significance of the LR model and the importance of XGBoost. The coefficient and feature significance of the LR model are summarized in Table 5. The Disaster Experience (Yes) and Awareness of Information (Yes) predictors are both significant and positively correlated with disaster preparedness. The results also show that the elderly are more likely to prepare for disasters. Income is another essential factor that affects disaster preparedness, with those earning a monthly income of over USD 4000 being more likely to prepare than those earning under USD 999. Gender and education backgrounds do not significantly affect people's perceptions of disaster preparedness based

on the LR results. In terms of race, Asian people are less likely to be prepared for disasters compared to white people, while the preparedness of African-American, American Indian, or Hawaiian people does not significantly differ from that of white people.

**Table 5.** Coefficients and significance of logistic predictors.

| | Coefficient | Std. Error | Z Value | p Value |
|---|---|---|---|---|
| Intercept | −2.327 | 0.343 | −6.790 | $1.09 \times 10^{-11}$ |
| Experience (Yes) | 0.941 | 0.130 | 7.242 | $4.44 \times 10^{-13}$ |
| Information (Yes) | 1.227 | 0.121 | 10.126 | $<2 \times 10^{-16}$ |
| Age | 2.100 | 0.271 | 7.746 | $9.45 \times 10^{-15}$ |
| Income (1000–3999) | 0.302 | 0.234 | 1.292 | 0.196 |
| Income (4000–9999) | 1.145 | 0.243 | 4.710 | $2.47 \times 10^{-6}$ |
| Income (Over 10,000) | 1.549 | 0.277 | 5.594 | $2.21 \times 10^{-8}$ |
| Gender (Men) | 0.169 | 0.119 | 1.415 | 0.157 |
| Education (High School) | −0.311 | 0.275 | −1.131 | 0.258 |
| Education (Vocational School) | −0.048 | 0.347 | −0.137 | 0.891 |
| Education (College) | −0.184 | 0.271 | −0.679 | 0.497 |
| Education (College Graduate) | −0.337 | 0.276 | −1.222 | 0.222 |
| Education (Post-Graduate) | −0.526 | 0.294 | −1.787 | 0.074 |
| Ethnicity (African-American) | −0.222 | 0.145 | −1.535 | 0.125 |
| Ethnicity (Asian) | −0.940 | 0.399 | −2.355 | 0.0185 |
| Ethnicity (American Indian) | 0.031 | 0.452 | 0.068 | 0.946 |
| Ethnicity (Hawaiian) | 0.121 | 0.613 | 0.197 | 0.844 |

The XGBoost model has been used to compute the importance of each predictor, and the results are depicted in Figure 2. The importance values of the predictors reveal that Awareness of Information has the highest Gain importance value, which is over 0.3, making it the most significant predictor. The predictors Age and Disaster Experience (Yes) have the second and third highest importance values, with 0.237 and 0.14, respectively. The importance of Income (4000–9999) and Income (Over 10,000) is also high, being close to 0.1. These predictors also have low P-Values in the LR results, which further supports their significance in predicting disaster preparedness. On the other hand, the importance of Ethnicity (African-American), Gender (Men), Ethnicity (Asian), Education Level (High School), and Income (1000–3999) is lower, with values close to 0.02. The importance of the remaining predictors does not exceed 0.001. It is worth noting that these predictors are still considered significant, but their contribution to predicting disaster preparedness is relatively lower compared to the other predictors.

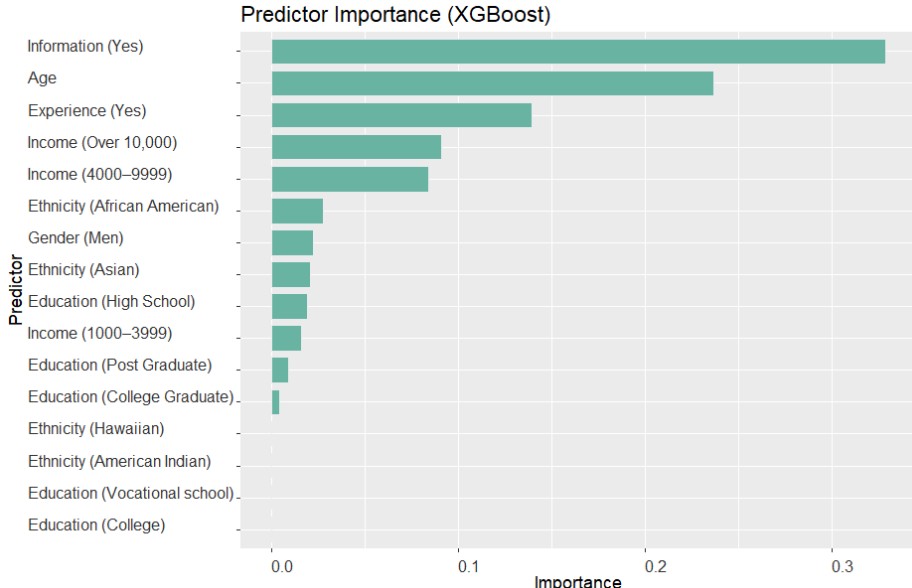

**Figure 2.** Predictors' importance.

The results from the XGBoost model and the LR model exhibit some similar insights in terms of feature importance. Out of the six predictors that show significance in the LR

model, five of them (Awareness of Information (Yes), Age, Disaster Experience (Yes), Income (4000–9999), and Income (Over 10,000)) also demonstrate relatively higher importance values in the XGBoost model.

The LR model indicates that if a household member has gained information about disaster preparedness, it is positively associated with preparedness. This finding is consistent with the XGBoost model, which shows that the Awareness of Information predictor has the highest importance value among all the predictors. These results suggest that FEMA's public awareness campaign, called "Ready", which provides disaster preparedness information through various channels such as television and websites, is an effective strategy to increase preparedness. In addition, the LR model also shows that disaster experience is positively associated with preparedness, which is consistent with the XGBoost model, where the Disaster Experience (Yes) predictor is the third most important feature. This indicates that individuals who have previously experienced disasters are more likely to be prepared for future disasters.

According to the LR results, age is positively associated with disaster preparedness, and it is the second most important feature in the XGBoost model. This suggests that older individuals are more likely to engage in disaster preparedness activities. One possible explanation for this trend is that the survey data show that older people are more likely to have received information about disaster preparedness than younger people. Specifically, the survey results indicate that 70% of people aged 65 years or older have learned about disaster preparedness, whereas only 43% of individuals surveyed by FEMA-NHS [31] reported having gained knowledge on the subject. This finding suggests that targeted campaigns to provide information and resources for disaster preparedness to younger age groups could be beneficial in improving overall disaster preparedness rates.

According to the XGBoost model, Income (4000–9999) and Income (Over 10,000) are the fourth and fifth most important variables, respectively. The LR model also indicates that both of these income brackets have a positive relationship with disaster preparedness. These findings align with our previous analysis, which showed that middle- and high-income households are more likely to engage in disaster preparedness activities compared to those with lower incomes. This may be because higher-income households have more resources and are better able to afford emergency supplies, insurance, and other preparations. Additionally, they may have more education and awareness of the importance of disaster preparedness.

Although ethnicity (Asian) is a significant predictor in the LR model, it is not a variable of top importance in the XGBoost model. This may be because the surveyed Asian people are generally younger and have less disaster experience. Specifically, the training dataset shows that the average age of the Asian interviewees is 36.53, which is much younger than the average age of the other ethnic groups. Additionally, the percentage of surveyed Asian people who have disaster experience is 58.8%, which is lower than the percentage of white people who have experienced a disaster, which is 73.2%. These factors could contribute to the lower importance value of ethnicity (Asian) in the XGBoost model, indicating that it may have a weaker relationship with disaster preparedness compared to other predictors.

### 3.4. Preparedness Level Prediction

In this section, we use Miami-Dade County (FL) as an example to predict its preparedness level based on 1000 simulated datasets, each containing 1000 simulated residents with demographic features and disaster-related features. The demographic features include Age, Income, Gender, Education Level, and Ethnicity, which are simulated based on the distribution obtained from the *2019 American Community Survey Single-Year Estimates* [38]. The probabilities of whether a person has a disaster experience and awareness of information are set to 0.72 and 0.43, respectively, which are observed from the FEMA NHS 2018 [31].

The density plot of the predicted disaster preparedness level by the LR model is presented in Figure 3. The blue dashed line in the figure represents the overall preparedness level (63.5%) of Miami-Dade County, as observed from the FEMA NHS 2019 dataset [13].

The red dashed line interval indicates the 95% confidence interval (CI) of the predicted preparedness level. The histogram is symmetric and has a peak at 0.645, which is the percentage of preparedness. The 95% CI ranges from 0.63 to 0.66, covering the Miami-Dade preparedness level from the real survey data. However, the observed preparedness level is not close to the axis of symmetry and is on the left-hand side. This indicates that the predicted preparedness level is higher than the surveyed value in most scenarios. Therefore, the results of the LR model are more optimistic than the surveyed preparedness level in Miami-Dade County. Figure 4 displays the preparedness level density obtained by the XGBoost model, as a comparison benchmark. The histogram of XGBoost is also symmetric, with a peak near 0.63, which is smaller than that of the LR model. The 95% CI ranges from 0.615 to 0.65, containing the surveyed Miami-Dade preparedness level. Unlike the LR model, the surveyed preparedness level is very close to the axis of symmetry in XGBoost. Based on the plot, the proposed framework using XGBoost shows better ability to obtain predicted results that are closer to the surveyed value in Miami-Dade County. This result indicates that the XGBoost model is more accurate in predicting the preparedness level than the LR model.

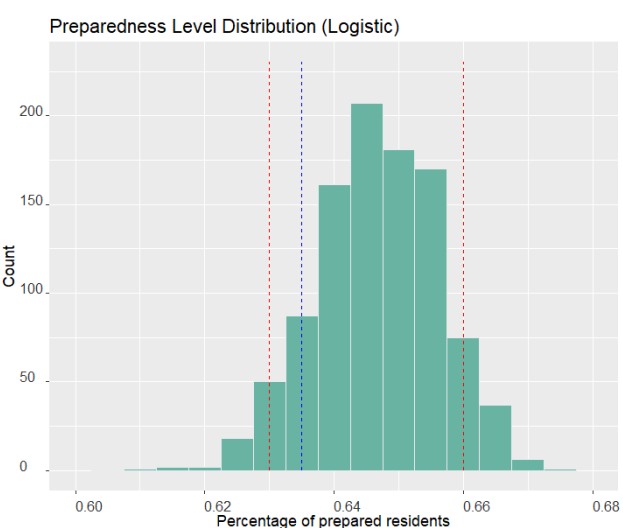

**Figure 3.** Predicted preparedness level (LR).

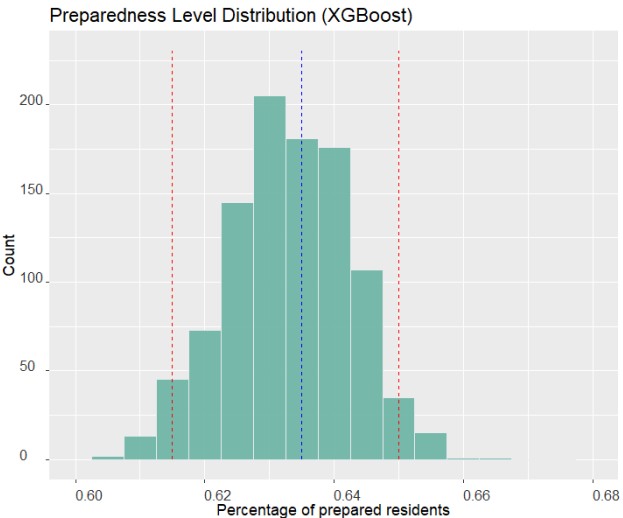

**Figure 4.** Predicted preparedness level (XGBoost).

The density plots obtained by LR and XGBoost for Miami-Dade County show that the results of XGBoost are closer to the surveyed preparedness level. One reason for this is that

the distribution of some demographic factors in the training dataset may differ from the actual distribution in the Miami-Dade County population. For instance, Table 5 shows that all other ethnicities, except white people, have negative coefficients, indicating that they are less likely to conduct disaster preparedness. However, around 75% of Miami-Dade residents are white people, according to the 2019 American Community Survey Single-Year Estimates [38]. This difference in the distribution of the ethnicity factor may lead to the predicted preparedness level by the LR model being higher than the actual level. In contrast, XGBoost has identified the top five predictors that have the most significant power to affect the preparedness level. The summation of the importance values of these five predictors is about 0.88, which means that these factors are more critical for the prediction of the preparedness level than the other factors. The ethnicity factor is not among the top five predictors in the XGBoost model, which may explain why the preparedness level predicted by XGBoost is closer to the surveyed value than LR.

*3.5. Sensitivity Analysis for Public Information Awareness*

In the context of disaster management, public information awareness about disaster preparedness refers to the extent to which individuals have knowledge and understanding of how to prepare for, respond to, and recover from natural disasters. The level of public information awareness about disaster preparedness can vary based on several factors, such as the frequency and type of disasters in a given region, the quality and accessibility of information provided by government agencies and other organizations, and individual attitudes and beliefs about the likelihood and impact of disasters. Effective communication and education initiatives can play a vital role in enhancing the level of public information awareness about disaster preparedness. By providing accessible and reliable information through multiple channels, including social media, television, and websites, individuals can become better equipped to handle the consequences of disasters. Moreover, individuals with higher levels of public information awareness about disaster preparedness are better able to make informed decisions and take appropriate actions during and after disasters. This, in turn, can help reduce the negative impact of disasters on individuals, families, and communities.

We next perform a sensitivity analysis on the awareness of information variable to investigate the impact of information awareness on disaster preparedness. The sensitivity analysis allows relief agencies to determine the effectiveness of their education and communication efforts in increasing public awareness about disaster preparedness. Specifically, it can help agencies identify which communication methods are most effective and which areas require more attention to improve public awareness. The results of the sensitivity analysis can also be used to justify the allocation of resources to disaster preparedness education and communication efforts.

Figures 5 and 6 illustrate how the disaster preparedness level changes when the percentage of residents who are aware of disaster preparedness information varies from 38% to 48%. In the figure, the density curves of the number of disaster preparedness when assuming 38%, 43%, and 48% of residents can effectively receive disaster preparedness information are represented by the gray, green, and yellow curves, respectively. Figure 5 displays the changes in disaster preparedness when varying the percentage of residents who receive effective disaster preparedness information using logistic regression in the proposed framework. Similarly, Figure 6 shows the changes in disaster preparedness when varying the percentage of residents who receive effective disaster preparedness information using XGBoost in the proposed framework. As the percentage of information awareness decreases from 43% to 38%, both LR and XGBoost models show a leftward shift in the density curve, indicating a decrease in the disaster preparedness level. Conversely, when the percentage increases from 43% to 48%, the density curve shifts to the right, indicating an increase in the disaster preparedness level. These results suggest that both LR and XGBoost models are sensitive to changes in the percentage of residents who have awareness of disaster preparedness information. Small increases in this percentage can lead to significant

improvements in disaster preparedness levels, which can be a compelling argument for investing in education and communication efforts.

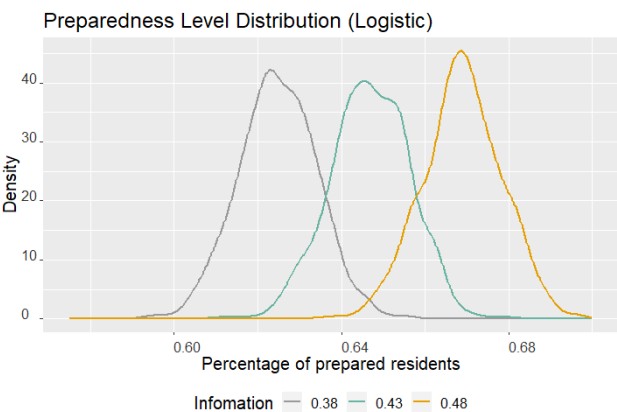

**Figure 5.** Predicted preparedness level comparison (LR).

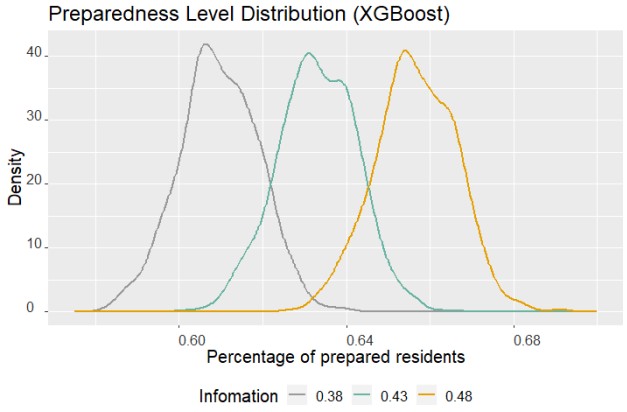

**Figure 6.** Predicted preparedness level comparison (XGBoost).

## 4. Conclusions

In this study, we propose a framework that combines machine learning models and simulation approaches to predict the level of disaster preparedness at the county level. Our framework uses 2017–2019 FEMA NHS data for model training and testing and simulates demographic features based on 2019 census data of Miami-Dade County. The results show that both the LR and XGBoost models can predict the level of disaster preparedness in Miami-Dade County, but XGBoost is more accurate than the LR model. The importance analysis of the predictors conducted in our study reveals that awareness of information, age, disaster experience, and income are significant predictors of disaster preparedness. These findings can inform public policymakers and emergency managers about the factors that influence the preparedness level of the population and can guide the development of targeted intervention programs.

Our framework provides a novel and effective approach for predicting disaster preparedness levels in a specific geographical area. It can help local authorities identify areas that are most vulnerable to disasters and design strategies to improve their level of preparedness. We believe that our framework can be useful for disaster management agencies and emergency responders in making more informed decisions and taking proactive measures to prevent or mitigate the impact of disasters. Future research can further refine the framework by incorporating additional demographic factors and expanding the geographic scope of the study to other regions.

Our study is limited by the accuracy of the survey results, as any inaccuracies can significantly impact our predictive results. The framework we propose relies on existing

survey data and uses machine learning to explore the relationship between disaster preparedness and demographic features, disaster experience, and information accessibility. By predicting the disaster preparedness of residents in affected areas, we hope to facilitate relief efforts in times of crisis. However, inaccurate data used during model training can hinder our ability to accurately determine these relationships through machine learning, leading to significant discrepancies between our predicted results and the actual preparedness of residents. This is a crucial limitation to consider, as relief operations based on incorrect information may fail to reach people in urgent need, exacerbating their suffering.

One direction for future study is to consider the integration of cutting-edge technologies, such as combining Blockchain and IoT, to incorporate real-time data sources for the construction of a disaster preparedness prediction framework. In their study, Ref. [39] demonstrates that implementing Blockchain alongside IoT in humanitarian aid supply chains results in location awareness, self-reporting, auto-correcting, and interoperability. Building on this, we plan to employ Blockchain and IoT in the future to track individual preparedness statuses in real-time. This crucial disaster preparedness information can then be securely stored and managed on a Blockchain, supplying the proposed framework with dependable, up-to-the-minute data to enhance prediction accuracy. Another promising direction for future research entails harnessing advanced technologies to augment the prediction accuracy of machine learning models within the proposed framework. This concept is inspired by the work of [40,41]. In [40], the authors employed Fuzzy Inference Systems (FIS) to tackle a personnel assignment problem, illustrating that FIS can adeptly accommodate the inherent uncertainty and imprecision related to human behavior and subjective evaluations. In [41], the authors applied FIS to a disaster relief volunteer management problem, using FIS to encapsulate decision-makers' knowledge and emulate the human reasoning process. By incorporating FIS into our framework, we aim to capture the nuances of public attitudes and preparedness levels more effectively, thus enhancing the accuracy of our predictions. Moreover, FIS enables the integration of expert knowledge and qualitative data, enriching our understanding of factors influencing disaster preparedness. This fusion could ultimately result in more targeted and efficient interventions by relief agencies, strengthening community resilience in the face of catastrophic events.

**Supplementary Materials:** The following supporting information can be downloaded at: https://www.mdpi.com/article/10.3390/su15108035/s1, Table S1: Parameter setting.

**Author Contributions:** Conceptualization, Z.J., Y.C., W.J. and R.J.; methodology, Z.J., Y.C., W.J. and R.J.; software, Z.J. and T.-Y.Y.; validation, Z.J. and T.-Y.Y.; formal analysis, Z.J.; data curation, Z.J.; writing—original draft preparation, Z.J.; writing—review and editing, Z.J., Y.C., W.J., Z.D. and R.J.; visualization, Z.J.; supervision, R.J.; project administration, R.J. All authors have read and agreed to the published version of the manuscript.

**Funding:** This research received no external funding.

**Institutional Review Board Statement:** Not applicable.

**Informed Consent Statement:** Not applicable.

**Data Availability Statement:** Publicly available datasets were analyzed in this study. These data can be found here: https://www.fema.gov/about/openfema/data-sets/national-household-survey (accessed on 1 May 2022).

**Conflicts of Interest:** The authors declare no conflict of interest.

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
