# Peer review of "Leveraging Machine Learning and Simulation to Advance Disaster Preparedness Assessments through FEMA National Household Survey Data"

_sustainability, doi:10.3390/su15108035_

Round 1
Reviewer 1 Report
This is a timely effort on "Predicting Household Disaster Preparedness: An Integrated Machine Learning and Simulation Framework". However, there are few suggestions to improve this manuscript.
1. Novelty needs to be highlighted as compared to already accomplished work in the area of current study.
2. Number of references seems to be less than 30. These should be increased up to 35 with minding the most relevant ones.
3. What does ROC stands for?
4. Please define the abbreviations first before using them in the manuscript e.g. CV.
5. Instead of mentioning the table in the appendix, please put these tables or figures in the supplementary file.
6. What was the accuracy of the framework used to predict the disaster preparedness in this study?
7. The discussion on Figures 5 and 6 are not clear.
8. The title must have been provided with the premises for which the data were taken and simulated.
Reviewer 2 Report
In this study, the authors tried to develop a framework that integrates machine learning and simulation, to enhance the comprehensiveness of disaster preparedness assessments.
The application of machine learning methods to household disaster preparedness is a novel attempt in traditional studies on disaster preparedness based on questionnaires and interviews.
Overall, the content organization of the manuscript is complete, and the logic is clear. The conclusion from this study is reasonable. There are only a few issues that the authors is recommended to consider further.
1. The disaster preparedness variable only uses a binary question, which may not fully reflect the households' disaster preparedness attitude and ability. It is recommended to increase the response variables for disaster preparedness when possible.
2. Why do training sets use data sets from 2017 and 2018 NHS, while testing sets use data sets from 2019 NHS? Is it appropriate to select a certain proportion of data as a test set after mixing?
3. It is suggested to discuss the limitations of the approaches used in this study.
Reviewer 3 Report
1. There are already machine learning approaches regarding disaster management, including preparedness:
· https://www.mdpi.com/2504-4990/4/2/20
· https://www.sciencedirect.com/science/article/abs/pii/S2212420922004952
2. Rather than coupling simulation and machine learninig, the authors are just resorting to monte carlo simulation to generate a dataset for feeding the neural network based on the variable’s distribution. But this is not simulation, just a procedure to extract deterministic realizations from stochastic variables.
3. I cannot see the novelty nor the contribution of the research, I think that the authors should come back to the drawing board with their research.
Round 2
Reviewer 1 Report
The authors have now improved the manuscript. The Manuscript can be accepted in current form.
Author Response
We thank the reviewer for the valuable feedback on the paper throughout the review process, and for accepting the paper.